High expression of SLC26A6 in the kidney may contribute to renal calcification via an SLC26A6-dependent mechanism

http://orcid.org/0000-0002-6307-2859 Jiang Hongyang 1 2
Pokhrel Gaurab 1 2
Chen Yinwei 1 2
Wang Tao 1 2
Yin Chunping 3 cpyin888@163.com
Liu Jihong 1 2
Wang Shaogang 1 2
Liu Zhuo 1 2 tjmulz@163.com
1 Department of Urology, Tongji Hospital, Tongji Medical College, Huazhong University of Science and Technology , Wuhan , China
2 Institute of Urology, Tongji Hospital, Tongji Medical College, Huazhong University of Science and Technology , Wuhan , China
3 School of Pharmacy, Tongji Medical College, Huazhong University of Science and Technology , Wuhan , China
de sousa Maria
Electronic publication date: 2018 Jul 3
Publication date: 2018
Volume: 6
Electronic Location ID: e5192
Received 2018 Feb 23; Accepted 2018 Jun 18
Copyright: © 2018 Jiang et al.
Copyright year: 2018
Copyright holder: Jiang et al.
License: This is an open access article distributed under the terms of the Creative Commons Attribution License, which permits unrestricted use, distribution, reproduction and adaptation in any medium and for any purpose provided that it is properly attributed. For attribution, the original author(s), title, publication source (PeerJ) and either DOI or URL of the article must be cited.
License URL: https://creativecommons.org/licenses/by/4.0/

Keywords: SLC26A6, Oxalate, Lentivirus, Hyperoxaluria, Urolithiasis

Funding: National Natural Science Foundation of China 81400706 The work was funded by the National Natural Science Foundation of China (grant no. 81400706). The funders had no role in study design, data collection and analysis, decision to publish, or preparation of the manuscript.

==============================
Background

Solute-linked carrier 26 gene family 6 (SLC26A6), which is mainly expressed in intestines and kidneys, is a multifunctional anion transporter crucial in the transport of oxalate anions. This study aimed to investigate the role of kidney SLC26A6 in urolithiasis.

Methods

Patients were divided into two groups: stone formers and nonstone formers. Samples were collected from patients following nephrectomy. Lentivirus with Slc26a6 (lentivirus-Slc26a6) sequence and lentivirus with siRNA-Slc26a6 (lentivirus-siRNA-Slc26a6) sequence were transfected into rats’ kidneys respectively and Slc26a6 expression was detected using Western blot and immunohistochemical analyses. After administering ethylene glycol, oxalate concentration and prevalence of stone formation between the transgenic and control groups were measured using 24-h urine analysis and Von Kossa staining, respectively.

Results

Immunohistochemical and Western blot analyses indicated that stone formers had a significantly higher level of expression of SLC26A6 in the kidney compared with the control group. After lentivirus infection, the urinary oxalate concentration and rate of stone formation in lentivirus-Slc26a6-tranfected rats increased remarkably, while lentivirus-siRNA-Slc26a6-transfected rats showed few crystals.

Conclusion

The results showed that high expression levels of renal SLC26A6 may account for kidney stone formation. Downregulating the expression of SLC26A6 in the kidney may be a potential therapeutic target to prevent or treat urolithiasis.

Introduction

Nephrolithiasis is one of the most common urological conditions. An increase in both prevalence and incidence of the disease has been observed over the last several decades (Wang et al., 2017). Recurrence is a significant cause of the morbidity of the disease, causing a significant burden on the health care system (Khan, 2018). Calcium oxalate is the most prevalent type of kidney stone. A major risk factor for kidney stone formation is hyperoxaluria, which leads to an increase in urinary saturation of calcium oxalate and formation of calcium oxalate stones (Robertson & Peacock, 1980; Noori et al., 2014). Hyperoxaluria is caused mainly by the oxalate metabolism defect in the intestines and kidneys (Agrawal, 2017).

The Solute-linked carrier 26 (Slc26) gene family encodes anion exchangers capable of transporting different monovalent and divalent anions, including oxalate, Cl−, HCO3−, and sulfate (Aronson & Giebisch, 1997; Ko et al., 2002; Ohana et al., 2013). Oxalate is the end product of metabolism that must be excreted or sequestered (Marengo & Romani, 2008). SLC26A6, A1, and A7 are the three major anion transporters that mediate oxalate excretion and absorption causing stone formation (Burckhardt & Burckhardt, 2003; Hatch & Freel, 2005; Voss et al., 2006). SLC26A6 is a multiple anion transporter expressed mainly in the apical membrane of the intestines and kidneys. SLC26A6 has remarkable tissue-specific properties. It mediates the exchange of a cluster of anions including Cl−, HCO3−, sulfate, and oxalate, and is associated with oxalate transportation (Chernova et al., 2005; Clark et al., 2008). In the intestine, oxalate homeostasis is regulated mainly by SLC26A6-mediated oxalate secretion, imbalance of which results in hyperoxaluria and hyperoxalemia (Freel et al., 2006; Aronson, 2010). Knauf et al. (2011) demonstrated that intestinal oxalate secretion depends on an SLC26A6-dependent transcellular mechanism, while the absorption of oxalate takes place via a paracellular channel (Mount & Romero, 2004). Thus, enhancement in enteric oxalate secretion remains a therapeutic target, especially because duodenal oxalate absorption is entirely paracellular. However, 90–95% of endogenous oxalate is excreted by kidneys, and excess oxalate secretion may cause a high concentration of urinary oxalate (Prenen et al., 1981; Boer et al., 1985; Brzica et al., 2013). SLC26A6 not only mediates the secretion of oxalate in the intestine but also powers the gradient for transmembrane movement, including secretion and absorption in the kidney (Knauf et al., 2011; Alper & Sharma, 2013). This study hypothesized that the expression of SLC26A6 in the kidney would account for the prevalence of stones by affecting oxalate secretion.

Materials and Methods

Patients

Ethical statement

This study was performed at the Tongji Hospital, and the ethical approval was given by the Medical Ethics Committee at Tongji Hospital, Tongji Medical College, Huazhong University of Science and Technology (TJ-C20141225; Wuhan, China). All patients signed the informed consent form to participate. All methods were performed in full compliance with the Declaration of Helsinki.

Patients

Ten patients with calcium oxalate stones (stone group) and 10 patients with nonstone diseases, including tumor or tuberculosis (control group), between November 2015 and October 2016 in Tongji Hospital were included in this study. The patients in the stone group had undergone nephrectomy for severe hydronephrosis leading to the loss of renal function (Assimos et al., 2016). Clinical and laboratory characteristics of the patients are summarized in Table 1. The patients of calcium oxalate stones were selected according to previous records of hyperoxaluria and by the analysis of stones previously eliminated.

Table 1 Basic characteristics of patients.

Subject	Age (year)/Gender	Dx	Procedure	Stone analysis	
S1	58/M	Kidney stone (R)	Nephrectomy	CaC2O4, carbonate apatite	
S2	65/M	Kidney stone (R)	Nephrectomy	CaC2O4, carbonate apatite	
S3	60/F	Upper ureteral stone (L)	Nephrectomy	CaC2O4, carbonate apatite	
S4	59/F	Kidney stone (R)	Nephrectomy	CaC2O4, carbonate apatite	
S5	54/M	Kidney stone (R)	Nephrectomy	CaC2O4, carbonate apatite	
S6	54/M	Kidney stone (L)	Nephrectomy	CaC2O4, carbonate apatite	
S7	48/F	Kidney stone (R)	Nephrectomy	CaC2O4, carbonate apatite	
S8	61/F	Kidney stone (L)	Nephrectomy	CaC2O4, carbonate apatite	
S9	56/M	Kidney stone (R)	Nephrectomy	CaC2O4, carbonate apatite	
S10	49/M	Kidney stone (R)	Nephrectomy	CaC2O4, carbonate apatite	
C1	27/F	Renal tuberculosis (R)	Nephrectomy	NA	
C2	69/F	Kidney tumor (R)	Nephrectomy	NA	
C3	43/M	Kidney tumor (R)	Nephrectomy	NA	
C4	49/F	Renal tuberculosis (L)	Nephrectomy	NA	
C5	76/F	Kidney tumor (R)	Nephrectomy	NA	
C6	58/F	Renal tuberculosis (L)	Nephrectomy	NA	
C7	54/M	Kidney tumor (R)	Nephrectomy	NA	
C8	68/F	Kidney tumor (R)	Nephrectomy	NA	
C9	61/M	Kidney tumor (R)	Nephrectomy	NA	
C10	43/M	Kidney tumor (L)	Nephrectomy	NA	
Note:

S, stone former group; C, control group; F, female; M, male; Dx, diagnosis; R, right; L, left; NA, none.

Analysis of stone components

About 1.5 mg of stones were taken out and ground into powder. It was then mixed with 250 mg potassium bromide. This mixture was dried and subjected to a pressure of 20 MPa to make tablets with a thickness of 0.3–0.5 mm. Then, the spectral characteristic peaks of tablets were measured using an LIIR20 infrared spectrum analyzer (Lambda, Tianjin, China). The components were evaluated by comparing with the standard image.

Analysis of 24-h urine components of the stone and control groups

The standard protocol was followed to collect 24-h urine collected from the patients. After measuring the pH of each sample, 500 μL of 6 mol/L hydrochloric acid was added into each 10 mL fresh urine sample (Wu et al., 2015). Ions levels were measured by ion chromatography (883 Basic IC plus; Metrohm AG, Herisau, Switzerland) according to the methods described in a previous study (Wu et al., 2015). After high speed centrifugation by pretreatment column and 0.25 μm filter, samples were to be added into chromatographic column and analyzed the ion electrical conductance of samples (Chen et al., 2013).

Collection of kidney tissue samples

Renal cortex tissue was obtained using sterilized scissors following nephrectomy. Kidney tissue far away from a primary lesion was chosen in patients with tuberculosis and tumors who underwent nephrectomy and served as the control group. The tissues were stored at –80 °C and fixed with 4% paraformaldehyde (Boster, Wuhan, China) at room temperature for further analysis.

Western blot and immunohistochemical assays to detect the expression of SLC26A6 in the kidney

Samples were dissociated using radio immunoprecipitation assay and phenylmethanesulfonyl fluoride. The protein (40 µg/lane) was electrophoresed on 10% sodium dodecyl sulfate–polyacrylamide gels and transferred onto polyvinylidene fluoride membranes (Immobilon-P Transfer Membrane; Millipore Corporation, Burlington, MA, USA). The protein samples were equally mixed, and the expression of SLC26A6 in the two groups was analyzed. The primary antibody was goat anti-SLC26A6 (1:200; Santa Cruz Biotechnology, Santa Cruz, CA, USA) or mouse β-actin (1:500; Boster, Wuhan, China). The secondary antibody was rabbit anti-goat (1:5000; Boster, Wuhan, China) or goat anti-mouse (1:5000; Boster, Wuhan, China). After incubation with the secondary antibody at room temperature for 2 h, proteins were detected with the Bio-Rad Clarity Western Enhanced Chemiluminescence (ECL) Substrate (Bio-Rad Laboratories, Hercules, CA, USA) and ECL detection system (1705061; Bio-Rad Laboratories, Hercules, CA, USA). The immunohistochemical (IHC) assays were performed according to the standard protocol, and the dilution rate of anti-SLC26A6 primary antibody was 1:100. Microscopy (BX53; Olympus, Tokyo, Japan) was used to observe the expression of SLC26A6. Fluorescence intensities were measured using ImageJ software (National Institutes of Health, Bethesda, MD, USA).

Animal study

Ethical statement

The experimental protocol was conducted in accordance with the institutional ethical committee of Tongji Hospital, Tongji Medical College, Huazhong University of Science and Technology according to the “Guidelines for Experimental Animal Ethical Committee of Huazhong University of Science and Technology.” This study was approved by the Ethical Committee (TJ-A 20141219).

Animal model

A total of 40 male Sprague–Dawley rats (275–300 g) were obtained from the Animal Center of Tongji Medical College, Huazhong University of Science and Technology. They were reacclimatized to 12-h light/dark cycles at 23 °C for one week prior to the start of experiments in a specific pathogen-free animal house with a relative humidity of 45–55%. They were maintained on a diet consisting of standard laboratory chow and had free access to food. They were randomly divided into four groups (10 rats in each group): negative control, lentivirus (lv-SLC26A6), siRNA-lentivirus (siRNA-SLC26A6), and vector groups.

Lentivirus preparation

The Slc26a6 sequence (NCBI Gene ID: 301010; National Center for Biotechnology Information, Rockville, MD, USA) was selected according to the NCBI GeneBank. To knockdown rat SLC26A6, lentiviral vector piLenti-siRNA-red fluorescent protein (RFP)-based shRNA against Slc26a6 were constructed. The targeting sequence for shSlc26a6 is 5′-GGGAACTACTCAAGCTAAT-3′. The sequences were synthesized and inserted into the vector pWSLV as pWSLV-05-Slc26a6 and pilentivirus as piLenti-siRNA (Slc26a6)-RFP. Recombinant lentiviruses were packaged in HEK293T cells by co-transfection of Slc26a6-encoding plasmids and helper plasmids, such as pRSV-REV, pMDLg, and pVSV-G, with lipofectamine 2000 in proportion with 1:2:1:1, and the titer of lentivirus was calculated by gradually dilute the concentrated raw virus until observe just one fluorescence in cells. Then, HEK293T cells were seeded in 10-cm dishes until they reached a confluence of 80–90%. The infection complex was directly added to each dish. The cells were then incubated in a 5% CO2 incubator at 37 °C overnight. Lentivirus particle–containing culture medium was harvested 48 h post infection. The final titers of the lentiviral vector contained 109 Tu/mL.

Lentivirus infection into the kidneys of rats

The rats were placed in a prone position and anesthetized with sodium pentobarbital (40 mg/kg). An amount of 2 × 107 Tu per rat lentivirus were applied to subcapsular renal injection (2–3 points per kidney). Lentivirus subcapsular renal infection was successfully accomplished according to the previously published protocol (Zmuda, Powell & Hai, 2011). Every effort was made to minimize suffering.

Identification of infection using frozen sections and expression of Slc26a6 in transgenic rats

A total of two weeks after the successful infection of lentivirus into the kidneys of rats, the kidneys and duodenal segments of four rats from different groups (n = 4/group) were harvested under anesthesia. The tissues were instantly frozen in liquid nitrogen and stored at −80 °C. After embedding and routine processing, serial sections were observed under a fluorescent microscope to identify the fluorescent protein. Then IHC and Western blot assays were performed as described in the “Patients” section to measure the expression of Slc26a6 protein.

Crystal deposition and urinary oxalate in the kidneys of rats

Ethylene glycol (EG) was dissolved in water to titrate 1% EG. The remaining animals (n = 6/group) had free access to 1% EG. After 2 weeks of 1% EG treatment in all four groups, urine specimens from each group were collected at 24-h intervals using a metabolism cage. Ion chromatography (883 Basic IC plus; Metrohm AG, Herisau, Switzerland) was used to detect 24-h urinary oxalate according to the methods described in a previous study (Chen et al., 2013). The rats were killed under anesthesia, and kidneys were removed. Von Kossa staining was used to identify crystal formation in kidneys.

Statistical analysis

Data were presented as mean ± standard deviation. The statistical analysis was performed using two-way analysis of variance. Differences were considered significant if the P value was <0.05. All analyses were performed using SPSS version 22.0 (Statistical Software for Social Sciences, Inc., Chicago, IL, USA).

Results

Patient characteristics and urine biochemistry

From 2015 to 2016, a total of 20 subjects (10 stone formers and 10 nonstone formers) were recruited in this study (Table 1). The average age in both groups was 56.4 ± 5.32 vs 54.8 ± 14.76 years (stone formers vs nonstone formers). Stone formers included patients with kidney and upper ureteral stones, while the nonstone formers included patients suffering from renal tuberculosis or tumors. Stones were collected after nephrectomy, and subsequent component analysis of the stone was performed using a stone composition analyzer; CaC2O4 and carbonated apatite were detected as the main components.

After recruiting idiopathic stone formers, 24-h urine was collected and the urinary oxalate, calcium, citrate, magnesium, phosphorus, and pH were measured (Fig. 1A). The stone formers had high levels of urinary oxalate (65.72 ± 12.44 vs 31.63 ± 7.91 mg/24 h, P < 0.05), and calcium (382.61 ± 104.06 vs 226.89 ± 66.01 mg/24 h, P < 0.05), and low levels of urinary citrate (344.93 ± 57.78 vs 444.91 ± 65.26 mg/24 h, P < 0.05) compared with the control group. However, no significant difference in urinary magnesium (84.23 ± 9.91 vs 90.17 ± 9.35 mg/24 h, P = 0.18) and phosphorus (1,006.85 ± 466.87 vs 735.11 ± 164.87 mg/24 h, P = 0.10) was noted between the two groups. Urinary pH (6.25 ± 0.49 vs 6.30 ± 0.35, P = 0.79) showed no remarkable difference between the two groups.

Figure 1 24-h urine analysis and the expression of SLC26A6 in the kidneys of stone formers and nonstone formers.

(A) 24-h urinary analysis of stone formers and nonstone formers. n = 10. Compared with the nonstone formers, stone formers had a higher level of urinary oxalate, calcium and a lower urinary citrate. No significant difference was observed in urinary phosphorus, magnesium and pH between the two groups. (B, C) Western blot analysis of expression of SLC26A6 from kidney of stone formers and nonstone formers. Densitometry values were normalized to respective β-actin values before statistical analysis. n = 10. (D, E) One representative image for each group. A similar trend was seen in the IHC analysis (upper magnification: ×200, lower magnification: ×800). The asterisk (*) denotes glomerulus, and the arrow points to SLC26A6 protein. The mean percentage of SLC26A6 is shown in the column diagram. n = 10. The data are expressed as means ± standard deviation (SD); *P < 0.05.

Expression of SLC26A6 in patients with calcium oxalate stones

After successful nephrectomy, IHC and Western blot analyses were performed to detect the expression of SLC26A6 in the renal tissue (Figs. S1 and S2). Both IHC and Western blot assays showed significantly higher expression of renal SLC26A6 in the stone formers than in the control group (Figs. 1B–1E).

Lentivirus preparation and infection detection

Lentiviruses (Fig. 2A), namely pWSLV-05-Slc26a6 (lv-Slc26a6) and piLenti-siRNA (Slc26a6)-RFP (siRNA-Slc26a6), were marked with RFP and anti-puromycin gene. After subcapsular injection into the kidneys of rats for two weeks, the kidney and duodenal frozen sections were examined under a fluorescence microscope. Lentivirus infection was observed to be successful in the kidney of transfected groups compared with the control group (Fig. 2B).

Figure 2 Lentivirus preparation and transfection detection.

(A) The Slc26a6 sequence was inserted into a lentiviral vector as pWSLV-05-Slc26a6. And ds-siRNA anti-Slc26a6 sequence was inserted into a lentiviral vector as pLenti-siRNA (Slc26a6)-RFP. (B) After successful lentivirus insertion into the kidneys of rats, frozen sections of kidneys and duodenums were observed under a fluorescence microscope. Cell nucleus were stained with DAPI. RFP was used as a marker to show successful insertion. Compared with the control group, transfection in the kidneys of the experimental groups (lv-Slc26a6 and siRNA-Slc26a6 groups) was successful as seen under a fluorescence microscope (magnification ×200). No difference of RFP expression was observed in the duodenum tissue between the control group and the experimental groups. The data are expressed as means ± SD (n = 4 rats/group); *P < 0.05.

Comparison of the expression of Slc26a6 in the kidneys and duodenum of rats

Western blot and IHC analyses were used after successful lv-Slc26a6 and siRNA-Slc26a6 infection to demonstrate the variation in the expression of Slc26a6 in the kidneys and duodenum of rats (Figs. 3A–3D). No significant difference was observed in the expression of Slc26a6 among the Slc26a6-lentivirus, siRNA-Slc26a6, vector-lentivirus, and normal control groups in duodenal tissue, suggesting that lentivirus did not alter the expression of duodenal Slc26a6. Western blot and IHC analyses of renal tissue of lv-Slc26a6-transfected rats showed higher expression of Slc26a6 compared with that in the normal control groups. Transfecting siRNA-Slc26a6 decreased the expression of Slc26a6 in the kidneys, whereas transfecting vector-lentivirus did not show any significant change in the expression of Slc26a6.

Figure 3 Expression of Slc26a6 in the duodenal and renal tissues of rats among different groups.

(A) Western analysis of duodenal tissue showed no difference in the expression of SLC26A6 among lv-Slc26a6, siRNA-Slc26a6, control, and vector groups. The data are expressed as means ± SD (n = 4 rats/group). (B) Evaluation of the expression of renal Slc26a6. The lv-Slc26a6 group showed significantly higher expression compared with the control group, while the level reduced in the siRNA-Slc26a6 group. No significant difference was observed in the expression in the vector and control groups. The data are expressed as means ± SD (n = 4 rats/group); *P < 0.05. (C) Immunohistochemical (IHC) analysis was performed to detect the expression of duodenal SLC26A6 in different groups. No difference was observed in the expression of lv-Slc26a6 (I), siRNA-Slc26a6 (II), control (III), and vector groups (IV). The data are expressed as means ± SD (n = 4 rats/group). (D) IHC was performed to detect the expression of renal Slc26a6 in different groups (upper magnification: ×100, lower magnification: ×400). Lv-Slc26a6 (I) had significantly higher expression compared with the control group (III), whereas siRNA-Slc26a6 (II) had lower expression compared with the control group. No significant difference was observed between the vector (IV) and control groups (III). The data are expressed as means ± SD (n = 4 rats/group); *P < 0.05.

Rate of stone formation and urinary oxalate concentration in rats

The rate of stone formation and urinary oxalate concentration in rats with lv-Slc26a6 (Slc26a6 group) increased remarkably, whereas the rate in those with siRNA-Slc26a6 (siRNA group) decreased after lentivirus infection. The supplementation of drinking water with 1.0% EG induced hyperoxaluria, and the subsequent oxalate excretion was measured. The urinary oxalate concentration of rats was 71.90 ± 17.21, 32.80 ± 12.20, 47.73 ± 10.50, and 53.24 ± 14.97 µmol/24 h, in the lv-Slc26a6, siRNA-Slc26a6, control, and vector groups, respectively (Fig. 4A). Von Kossa staining was used to quantify the mineralization status. The result showed 11.77 ± 3.56/field, 5.83 ± 3.47/field, 9.00 ± 2.11/field, and 8.78 ± 3.34/field in the Slc26a6, siRNA, control, and vector groups, respectively (Figs. 4B and 4C).

Figure 4 Transfection with Slc26a6 or siRNA-Slc26a6 led to different urinary oxalate and crystal formation in rats.

(A) Supplementation of the drinking water with 1.0% EG induced hyperoxaluria; 24-h urine was collected by putting the rats in a metabolic cage. The oxalate level in urine was measured by ion chromatography, and the result showed that Slc26a6-lentivirus-transfected rats had significantly higher urinary oxalate excretion compared with the vector and control groups. siRNA-Slc26a6-transfected rats had less urinary oxalate compared with the control group. The data are expressed as means ± SD (n = 6 rats/group); *P < 0.05. (B) Von Kossa staining was used to observe the crystal formation in different groups. The Slc26a6-lentivirus group had increased crystal formation (B-I) compared with the normal control (B-III) and vector-lentivirus (B-IV) groups. The siRNA-lentivirus (B-II) group had the least crystal formation. Arrows point to crystals (magnification, ×200). (C) Crystal formation status in different groups. The Slc26a6-lentivirus group has the highest crystal formation compared with other groups, and the siRNA-Slc26a6 transgenic group had the least crystal formation. The data are expressed as means ± SD (n = 6 rats/group); *P < 0.05.

Discussion

Oxalate secretion in proximal tubules is associated with the expression of both basolateral SLC26A1 and apical SLC26A6 in the kidneys (Xie et al., 2002; Freel et al., 2006). Numerous studies have demonstrated a close relationship between the expression of SLC26A6 and kidney stone formation, but its exact role in the disease remains unknown (Hirata et al., 2012; Ohana et al., 2013; Landry et al., 2016; Lu et al., 2016). Jiang et al. (2006) demonstrated that Slc26a6−/− mice had a susceptibility to urolithiasis. The present case–control study examined the relationship between kidney stones and the role of SLC26A6 in stone formation.

Tissue samples were collected from patients with renal stones (stone former group) and patients suffering from renal tuberculosis or tumors (control group) to reveal the role of renal SLC26A6 in stone formers. Western blot and IHC analysis results showed a direct relationship between kidney stone formation and expression of SLC26A6. However, whether the increased expression of SLC26A6 in the stone group is the cause or result of the disease was difficult to predict.

Furthermore, infection of lentivirus with Slc26a6 sequence and lentivirus with siRNA anti-Slc26a6 into the kidneys of rats was achieved. The renal subcapsular injection technique was used to ensure the organ-specific expression of the virus and avoid expression in other organs especially in the intestinal tract (Zmuda, Powell & Hai, 2011). Figure 2B shows the successful infection of the virus and change in the expression of Slc26a6 specifically in the kidneys. An increase in oxalate excretion and crystal formation was noted with an increase in the expression of Slc26a6, and vice versa. A classical research demonstrated that intestinal oxalate secretion defect in Slc26a6-null mice could result in enhanced net absorption of oxalate leading to a high incidence of calcium oxalate urolithiasis (Jiang et al., 2006). Unlike the whole-body knockout, only the expression of renal Slc26a6 was regulated in the present study. Therefore, intestinal Slc26a6 that mediates oxalate excretion could function properly and not result in enhanced net absorption of oxalate. Besides, only a partial but significant defect in sulfate–oxalate exchange was observed compared with the complete loss of Cl−–oxalate exchange activity in Slc26a6-null mice (Jiang et al., 2006). This finding was consistent with the result of the present study that the concentration of oxalate was higher in rats with lv-Slc26a6 than in normal rats. Therefore, upregulated and downregulated renal Slc26a6 could lead to enhanced and reduced net excretion of oxalate that manifested as higher and lower urinary oxalate concentration, respectively (Fig. 5 shows the proposed mechanisms of oxalate transport across the renal epithelium in the proximal tubule) (Kleta, 2006). Moreover, the expression of Slc26a6 on renal tubular epithelial cells mediated influx or outflow of oxalate, leading to an increase in the intracellular oxalate content. Oxalate in the cell could stimulate the production of reactive oxygen species and inflammatory factors, resulting in cellular damage (Patel et al., 2006; Zhang et al., 2017). Injury in renal tubular epithelial cells (NRKs) is an initial mechanism for crystal nucleation, attachment, and retention, which was consistent with the result of the present study (Jonassen et al., 2003; Khan, 2004; Zuo et al., 2011; Abhishek et al., 2017).

Figure 5 Proposed mechanisms of oxalate transport across the renal epithelium in the proximal tubule.

In the renal proximal tubule cells, oxalate transport is associated with Slc26a1 expressed on basolateral membrane and Slc26a6 expressed on apical membrane. Slc26a1 mediates the uptake of oxalate in exchange for reabsorbed sulfate (or Cl− or HCO3−). The high expression of Slc26a6 mediates more secretion by oxalate–Cl− exchange and reabsorption by sulfate–oxalate exchange. According to the results, oxalate–Cl− exchange possessed a dominant position resulting in the enhanced net secretion of oxalate.

The animal experiment results showed that Slc26a6 was one of the causes of kidney stone formation. The potential mechanism might be that Slc26a6 led to the secretion and reabsorption of oxalate, resulting in higher oxalate concentration in urine and interstitium (Marengo & Romani, 2008). A high-oxalate concentration could lead to tissue inflammation and crystal deposition in a short time (Albert et al., 2017). Therefore, the lv-Slc26a6 group had the highest level of crystals and the siRNA-Slc26a6 group had the least level of crystals (Fig. 4C). Besides, only one time point (two weeks) for the crystal production was observed in this study. The crystals in the kidney might be slowly eliminated with time. Therefore, studies on setting up a time gradient are needed in the future (Okada et al., 2010; Yasui et al., 2014).

This study showed that the increased expression of renal Slc26a6 caused higher oxalate concentration in urine, leading to an increased crystal deposition. Moreover, reducing the expression of renal Slc26a6 by siRNA injection attenuated stone formation. However, the study had several limitations. First, due to the small cohort size, future studies with a larger number of patients are needed to confirm the results. Second, Slc26a6 was an important but not the only oxalate transporter; therefore, measuring oxalate flux mediated by other proteins and oxalate concentration in other areas help better understanding of oxalate transport. Third, all experimental procedures were subject to ethical approval. Patients with renal tuberculosis and cancer were selected as the control group because it was nearly impossible to acquire a sample of normal human renal tissue for this study. Moreover, the results of previous studies including samples from patients with cancer as the control group have been widely accepted (Chi et al., 2017). Hence, further studies with a large sample size are needed to address other similar mechanisms involved in calcium oxalate stone formation and validate the role of SLC26A6 in oxalate stone formation.

Conclusion

In conclusion, the present study demonstrated that the overexpression of Slc26a6 in the kidneys increased oxalate excretion and urinary oxalate concentration, contributing to an increase in the prevalence of stone formation. Downregulating the expression of SLC26A6 in the kidneys might be a potential therapeutic target to prevent or treat urolithiasis.

Supplemental Information

Supplemental Information 1 Raw data.

Click here for additional data file.

Supplemental Information 2 Uncropped blots of Figures.

Click here for additional data file.

Supplemental Information 3 Fig. S1. Immunohistochemistry assay for 10 stone formers (S1–S10).

Kidney tissues from stone formers were stained by IHC assay to detected SLC26A6 expression. The brown part represents the expression of SLC26A6 (magnification: X200).

Click here for additional data file.

Supplemental Information 4 Fig. S2. Immunohistochemistry assay for 10 non-stone formers (C1–C10).

Kidney tissues from non-stone formers were stained by IHC assay to detected SLC26A6 expression. The brown part represents the expression of SLC26A6 (magnification: X200). Compare to the Fig. S2, mean pixel of SLC26A6 was lower in non-stone former groups.

Click here for additional data file.

Additional Information and Declarations

Competing Interests

Author Contributions

Human Ethics

Animal Ethics

Data Availability

The authors declare that they have no competing interests.

Hongyang Jiang conceived and designed the experiments, performed the experiments, contributed reagents/materials/analysis tools, prepared figures and/or tables, authored or reviewed drafts of the paper, approved the final draft.

Gaurab Pokhrel prepared figures and/or tables, authored or reviewed drafts of the paper, approved the final draft.

Yinwei Chen performed the experiments, authored or reviewed drafts of the paper, approved the final draft.

Tao Wang performed the experiments, authored or reviewed drafts of the paper, approved the final draft.

Chunping Yin analyzed the data, authored or reviewed drafts of the paper, approved the final draft.

Jihong Liu analyzed the data, authored or reviewed drafts of the paper, approved the final draft.

Shaogang Wang analyzed the data, authored or reviewed drafts of the paper, approved the final draft.

Zhuo Liu conceived and designed the experiments, contributed reagents/materials/analysis tools, authored or reviewed drafts of the paper, approved the final draft.

The following information was supplied relating to ethical approvals (i.e., approving body and any reference numbers):

This study was performed at the Tongji Hospital, and the ethical approval was given by the Medical Ethics Committee at Tongji Hospital, Tongji Medical College, Huazhong University of Science and Technology (Wuhan, China; TJ-C20141225).

The following information was supplied relating to ethical approvals (i.e., approving body and any reference numbers):

The experimental protocol was conducted in accordance with the institutional ethical committee of Tongji Hospital, Tongji Medical College, Huazhong University of Science and Technology according to the “Guidelines for Experimental Animal Ethical Committee of Huazhong University of Science and Technology.”

The following information was supplied regarding data availability:

The raw data are provided in the Supplemental Files.

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
