# Peer review of "High expression of SLC26A6 in the kidney may contribute to renal calcification via an SLC26A6-dependent mechanism"

_PeerJ, doi:10.7717/peerj.5192_

## Round 0.1 · original submission · Major Revisions

Thank you for submitting your work for publication to PeerJ.

As per the appended comments, all reviewers felt that the paper needs major revisions before it can be considered for publication by PeerJ.

In addition to all the comments made by the referees you should address in greater depth the apparent contradiction between your conclusion and Jiang's et al., earlier results (2005) with Slc26a6-null mice discussed lightly in your ms.

Reviewer 1 ·

Basic reporting

1. In the introduction section, more background information is needed for the biological process and chemistry of kidney stone formation. Also, the authors can move the content about oxalate secretion from the discussion section to the introduction section.

2. References are needed for Lines 37, 38, 42, 44.

3. Bright-field images are needed for Figure 2b.

4. In Table 2, the authors should also provides the sequences of other primers and synthetic fragments.

5. In Line 80, write the full name of the acronym IHC. Also, write the acronym SDS-PAGE in Line 83.

6. In Line 91, in triplicates is more often used than thrice.

7. In Line 119 and other places, it is lentivirus ‘infection’ not ‘transfection’.

8. In all figures, y axis units are missing.

9. In Line 255, is it Figure 3b or 4c that should be referred to here?

Experimental design

1. In Line 74, the authors need to describe the method here other than citing a previous publication.

2. About lentiviral infection, which lentiviral infection system did the authors use and how was the titer calculated? More experimental details are expected, such as what are the packaging and envelope plasmids, transfection reagent.

Validity of the findings

In Figure 3a and 3b, it is strange that lentivirus infection did not change Slc26a6 gene expression in duodenal tissue. I am wondering if the infection was successful, maybe the duodenal tissue is hard to infect using the specific lentiviral infection system. Did the authors see RFP expression in the tissue? If so, the authors should provide the image and give a hypothesis of the tissue-specific expression.

Additional comments

The authors should check the usage of acronym. For example, in the legend of Figure 3, standard deviation and SD were used alternatively. The same problem happened to IHC, too.

Reviewer 2 ·

Basic reporting

The manuscript in title of High expression of SLC26A6 in the kidney may contribute to renal calcification via an SLC26A6 dependent mechanism aims to investigate the role of kidney SLC26A6 in urolithiasis. This is a nice manuscript and been well prepared, however, revision should be done before it could be accepted for publication.

More information should be added in the instruction section;

Experimental design

The experimental design is good and been well conducted, however, several issues need be fixed. The method for lentivirus transfection into the kidneys of rats is questionable; regarding the efficiency of lentivirus subcapsular renal transection. Please provide more information about this procedure.

Validity of the findings

1. Better images should be provided to demonstrate the location of SLC26A6
2. In line 101, please change the word “animal culture”.
3. Images in fig 1D are not comparable, please update them.
4. English editing is needed.

Additional comments

This is a nice manuscript and been well prepared, however, revision should be done.

·

Basic reporting

The paper is well constructed, the hypothesis is sound, te approach is innovative, the topic is relevant.
It is well written, the figures are OK
References OK

Experimental design

The experimental design is appropriate to the hypothesis
Methods are well described

Validity of the findings

Data are robust. The graphics should be changed according to my comments in the section below

Additional comments

All the 10 patients identified as stone formers had hyperoxaluria. Since there are other stone types, please explain if the patients were selected according to previous records of hyperoxaluria or by the analysis of stones previously eliminated.
In those 10 patients was a diet rich in oxalates excluded as a cause for the stone formation?
The transfection of lentivirus with Slc26a6 sequence and lentivirus with siRNA anti-226 Slc26a6 into the kidneys of rats was achieved by injection in the organs. How many injections were done per kidney to obtain a good spread of the vector?

For how long can the vectors remain in the kidney?

The fragments of kidney tissue used for Western blot and immunohistochemical staining were obtained from parts of the organ well apart from the local of the injections? Why two weeks was the time point decided and not 4 or 8?

90%–95% endogenous oxalate is excreted by kidneys. Did the animals in the siRNA-Slc26a6 groups have any evidence of oxalate overload in the organism?
Transfection with lv-Slc26a6 and siRNA-Slc26a6 was carried out in 4 plus 4 animals. This is a relatively low number. Please show individual values in fig 3 and 4 for each one of the transfected rats using a graphic plot as used in fig 1 for patients. This request is for valid for Western Blot, immunohistochemical staining intensity, urinary oxalate and crystal formation.

---

## Round 0.2 · accepted · Accept

Based upon the fact that the Authors responded to all the queries made and changed the manuscript and figures in a satisfactory way, the paper is now acceptable for PeerJ

I would like to thank the reviewers for their enormously helpful and careful work.

Reviewer 1 ·

Basic reporting

no comment

Experimental design

no comment

Validity of the findings

no comment

Additional comments

The revised version addresses all my concerns. I recommend the acceptance of this manuscript.

Reviewer 2 ·

Basic reporting

Authors made nice revision, no more comments.

Experimental design

Authors made nice revision, no more comments.

Validity of the findings

Authors made nice revision, no more comments.

Additional comments

Authors made nice revision, no more comments.

·

Basic reporting

Authors responded to all the queries made and changed the manuscript and figures in a satisfactory way

Experimental design

OK

Validity of the findings

OK

Additional comments

No more comments